# Infection control, occupational and public health measures including mRNA-based vaccination against SARS-CoV-2 infections to protect healthcare workers from variants of concern: A 14-month observational study using surveillance data

Annalee Yassi[1,2,3]*, Jennifer M. Grant[2,3,4,5], Karen Lockhart[1], Stephen Barker[1], Stacy Sprague[6], Arnold I. Okpani[1,2], Titus Wong[3,4,5], Patricia Daly[1,7], William Henderson[8,9], Stan Lubin[2], Chad Kim Sing[10,11]

1 School of Population and Public Health (SPPH), University of British Columbia (UBC), Vancouver, British Columbia (BC), Canada, 2 Physician Occupational Safety and Health, Vancouver Coastal Health (VCH), Vancouver, BC, Canada, 3 Department of Pathology and Laboratory Medicine, Vancouver Coastal Health (VCH), Vancouver, Canada, 4 Division of Medical Microbiology and Infection Prevention, VCH, Vancouver, Canada, 5 Division of Medical Microbiology, Department of Laboratory Medicine and Pathology, UBC, Vancouver, Canada, 6 Employee Safety, Health and Wellness, VCH, Vancouver, Canada, 7 Public Health, VCH, Vancouver, Canada, 8 Department of Medicine, UBC, Vancouver, Canada, 9 Vancouver Acute, Emergency Operations Centre, VCH, Vancouver, Canada, 10 Medicine, Quality and Safety, VCH, Vancouver, Canada, 11 Department of Emergency Medicine, UBC, Vancouver, Canada

* Annalee.Yassi@ubc.ca

## Abstract

### Background

We evaluated measures to protect healthcare workers (HCWs) in Vancouver, Canada, where variants of concern (VOC) went from <1% VOC in February 2021 to >92% in mid-May. Canada has amongst the longest periods between vaccine doses worldwide, despite Vancouver having the highest P.1 variant rate outside Brazil.

### Methods

With surveillance data since the pandemic began, we tracked laboratory-confirmed SARS-CoV-2 infections, positivity rates, and vaccine uptake in all 25,558 HCWs in Vancouver Coastal Health, by occupation and subsector, and compared to the general population. Cox regression modelling adjusted for age and calendar-time calculated vaccine effectiveness (VE) against SARS-CoV-2 in fully vaccinated ($\geq$ 7 days post-second dose), partially vaccinated infection (after 14 days) and unvaccinated HCWs; we also compared with unvaccinated community members of the same age-range.

### Findings

Only 3.3% of our HCWs became infected, mirroring community rates, with peak positivity of 9.1%, compared to 11.8% in the community. As vaccine coverage increased, SARS-CoV-2

**Data Availability Statement:** The data underlying the results presented in this study are available at https://innovation.ghrp.ubc.ca/covid19/data-2021.05.17.xlsx.

**Funding:** Funding was received from the Canadian Institutes of Health Research (CIHR) and the International Development Research Centre (IDRC) through the rapid research funding opportunity from CIHR under PI Yassi and grant, "Protecting healthcare workers from COVID-19: A comparative contextualized analysis." The work is also supported and funded by Doctors of BC and Vancouver Coastal Health. The funders had no role in study design, data collection and analysis, decision to publish, or preparation of the manuscript.

**Competing interests:** The authors have declared that no competing interests exist.

infections declined significantly in HCWs, despite a surge with predominantly VOC; unvaccinated HCWs had an infection rate of 1.3/10,000 person-days compared to 0.89 for HCWs post first dose, and 0.30 for fully vaccinated HCWs. VE compared to unvaccinated HCWs was 37.2% (95% CI: 16.6–52.7%) 14 days post-first dose, 79.2% (CI: 64.6–87.8%) 7 days post-second dose; one dose provided significant protection against infection until at least day 42. Compared with community infection rates, VE after one dose was 54.7% (CI: 44.8–62.9%); and 84.8% (CI: 75.2–90.7%) when fully vaccinated.

## Interpretation

Rigorous droplet-contact precautions with N95s for aerosol-generating procedures are effective in preventing occupational infection in HCWs, with one dose of mRNA vaccination further reducing infection risk despite VOC and transmissibility concerns. Delaying second doses to allow more widespread vaccination against severe disease, with strict public health, occupational health and infection control measures, has been effective in protecting the healthcare workforce.

## Introduction

Healthcare workers (HCWs) worldwide face occupational risk of infectious disease [1]. COVID-19 has highlighted this risk, particularly in the early stages of the pandemic when personal protective equipment (PPE) was lacking in many settings [2,3], compounding the mental health impacts of working on the frontline [4,5]. In many settings globally, HCWs have continued to provide patient care often in exhausting long shifts, and amidst fear of becoming infected and transmitting to family, friends, patients and co-workers, sometimes with new responsibilities and facing emotionally fraught decisions. Tragic deaths of medics and frontline healthcare workers continue to occur across the globe, despite the recognition that protecting the healthcare workforce is a prerequisite to the safety of patients and the health of the population at large.

Infection rates have been relatively low in the Canadian healthcare workforce [6] compared to elsewhere [7,8], with a positivity rate of 6.5% by September 2020, no higher than for the general Canadian population. Nonetheless, with growing concern about the possibility of greater airborne transmission [9,10], especially in the context of variants of concern (VOC), vigilance regarding protecting HCWs remains important in Canada, as it is worldwide [11]. While combinations of PPE and other non-pharmaceutical interventions are thought to be useful, there is increasing consensus that the most effective means of protecting HCWs is vaccination. Just how effective the various non-pharmaceutical interventions have been, and what the implications are for their ongoing application now that vaccines are available, are still topics of important debate. Moreover, while it is increasingly well-established that the commonly approved vaccines protect against severe illness [12], there has been no real-world data to date on the performance of vaccines against the P.1 variant, and the issues of how long a single dose of a two-series vaccine regiment remains protective against infection beckons further research.

HCWs were amongst the first groups to be vaccinated in the province of British Columbia (BC) in Canada, and thus serve not only as a population to be studied with respect to their own protection but also as a sentinel population to assess vaccine effectiveness, compared to the general population. During this period, the dominant variants changed from <1% VOC to

>92%, with the B1.1.7 and P.1 variants dominating; Vancouver, BC was documented at that time as having the highest rate of P.1 variant outside of Brazil [13].

This study therefore had two main objectives. First, we tracked the risk of COVID-19 infections in our cohort of HCWs compared to the general population since the beginning of the pandemic, examining risk by subsector and occupational group, to assess the effectiveness of the occupational precautions implemented to date. Second, we sought to examine the impact of the mRNA vaccine–including delaying the second dose–on COVID-19 infection in HCWs in a jurisdiction with high levels of the P.1 variant [13], reported to be 2.5-times more transmissible than the wild variant [14].

## Methods

### Setting and study population

British Columbia, as all Canadian provinces, offers universal healthcare coverage through a single-payer system, with all residents offered a Personal Health Number (PHN); non-permanent residents, including temporary foreign workers, refugees and undocumented immigrants are also able to obtain testing and vaccination free of charge, with a numeric identifier assigned to them for COVID-19 testing and vaccination purposes. Vancouver Coastal Health (VCH) and Fraser Health Authority (FHA) cover the greater Vancouver metropolitan area in BC. VCH provides all laboratory, community, hospital and long-term care (LTC) services to more than one million people. Public health measures during the pandemic included COVID-19 PCR test turnaround times of less than 24-hours, isolation within 24 hours of a positive test, and prompt isolation of close contacts for 10 days after symptom onset (20 days for hospitalized cases), as well as limits on travel, indoor activities and outdoor gatherings as needed to keep the caseload down and avoid pressure on the healthcare system.

Infection prevention and control (IPAC) measures starting March 29, 2020, required HCWs to wear a medical mask (ASTM level 1, 2 or 3), eye protection and gloves for all direct patient care, in addition to droplet and contact precautions when within 2 meters of COVID-19 suspect or confirmed patients. Use of an N95 or equivalent respirator was permitted based on a HCW's point-of-care risk assessment (PCRA) and was required when an aerosol generating medical procedure (AGMP) was performed on a positive or suspected COVID-19 patient. From November 4, 2020, all visitors and HCWs were required to wear a medical mask in all common areas. Cotton or non-approved masks were not permitted and double masking was strongly discouraged. There were no PPE disruptions during the pandemic, although extended use of facial PPE of up to 4 hours was encouraged. IPAC personnel ensured optimal administrative and engineering controls, ongoing staff instruction and rapid response to outbreaks. PPE measures were communicated in regular staff forums and bulletins plus targeted forums for medical staff.

In addition to the existing provincial occupational health and local Employee Health and Safety services, VCH established the Physicians Occupational Safety and Health (POSH) unit to service medical staff, providing prompt access to expert advice, as well as exposure notifications and assessments, and contact tracing for this often harder-to-reach group. POSH also conducts overall surveillance of HCW infection and vaccine rates within VCH, promoting vaccination and sending reminders to medical staff when eligible for vaccination with first or second dose.

Immunizations against COVID-19 began on December 15, 2020. Dose 1 was given first to LTC staff, residents and essential visitors, followed by highest risk acute HCWs (Emergency room, Intensive care unit and COVID medical unit staff) in late December 2020 and early 2021. Initially dose 2 was given 35 days after dose 1. The inter-dose interval was lengthened to

42 days in February 2020 then to 4 months (16 weeks) in early March 2021 in an effort to protect a greater number of people from severe disease and death [15] at a time of limited vaccine supply. By the end of the observation period for this study (May 13, 2021), almost all HCWs in this jurisdiction had been offered at least a single dose of vaccination, with some workers receiving 2 doses. Virtually all HCWs were vaccinated with either the Pfizer-BioNTech (93.3%) or Moderna (6.6%) COVID-19 vaccine (mRNA-1273). There were 310 HCWs (0.1%) who received AstraZeneca vaccine and they were excluded from our cohort. Vaccination of the general public started in February 2021, beginning with homeless and unstably housed, older age-groups and the clinically extremely vulnerable, as well as Indigenous nations, then essential workers and later high prevalence areas, working through the BC ethical framework [16].

Our HCW cohort includes all active healthcare employees (nurses, care aids/licensed practical nurses, allied health professionals, support staff, administrators, and other employees) as well as contracted medical staff (physicians, nurse practitioners, midwives, dentists, other medical staff) who worked in VCH between March 15, 2020 and May 13, 2021. Non-medical contractors (e.g. cleaning and food service staff) were not included in the database.

## Database and analysis

All COVID-19 vaccines provided in BC are recorded in a provincial database by PHN and other identifiers, regardless of immunization site. COVID-19 testing and results of PCR are updated daily, and used for prompt contact tracing and public health surveillance, in conjunction with the BC Centre for Disease Control (BCCDC). Data on HCW infection rates and vaccinations are also extracted daily to populate the occupational health database for HCWs, which includes their birthdate, sex, occupation and work location, among other variables such as respirator fit-testing results. Data extracted from this database are used for regular occupational health surveillance of all VCH HCW COVID-19 infections as well as monitoring and promoting vaccine uptake.

Community COVID-19 values and vaccination rates were collected from BCCDC, and the occupational health database was used for HCW infections and vaccination data. We plotted the COVID-19 rate in the VCH health workforce compared to the general population of similar age range, calculated over a moving 7-day period from March 1, 2020 to May 13, 2021. The population denominator was retrieved from Statistics Canada [17] grouped by age and health service regions. The data were merged, summarized and plotted using R (version 4.0.5). The combined data from VCH and FHA were used for community comparison for VCH HCWs, as VCH staff live in the larger Vancouver area which spans both health authorities.

To assess vaccine effectiveness against COVID-19 infections, all 25,558 VCH healthcare workers were classified according to vaccination status over the period of observation. For the 150-day interval from December 15, 2020 to May 13, 2021, each HCW had the period stratified into days unvaccinated, days vaccinated with one dose, and days vaccinated with two doses, allowing 14 or 7 days for vaccine effect for one or two doses respectively, to allow comparison with similar studies [18] assessing vaccine effectiveness. The 442 VCH healthcare workers who tested positive prior to December 15, 2020 were excluded from further analysis. A Cox regression model was fitted to the data adjusting for age and calendar time, and the proportional hazards assumption was verified.

Results were further assessed based on care sector and occupational categories. For the cumulative incidence plot, a HCW counted as positive the day that they tested positive or would be right-censored 14 days after the first dose or 7 days after the second dose if they did not test positive prior to the end of the period (May 13, 2021). The unvaccinated classification

had an origin time (t = 0) of December 15, 2020; the one dose classification had an origin time 14 days after the first dose; and the two-dose classification had an origin time 7 days after the second dose. A log-rank test was performed to test whether there was a significant difference between the incidence curves.

Ethical approval was provided by the Behavioural Ethics Review Board at the University of British Columbia under certificate H21-01138. This work was secondary analysis of occupational health data in which all information was anonymized before being extracted for analysis; individual consent was not required.

## Results

The rates of positive COVID-19 PCR tests per 100,000 population are shown by date in Fig 1, along with major points of interventions to protect HCWs. It can be seen that other than very early in the pandemic, before PPE guidance was provided and widely implemented, infection rates in HCWs paralleled those of the population at large, dramatically decreasing below that of the community at large as vaccination of HCWs was quickly rolled-out at a faster pace than in the general population. As the third surge, driven in part by a high proportion of the P.1 variant, was quickly brought under control through public health interventions including more widespread vaccination coverage of the population, the community rates fell, but HCW rates still remained well below those of the background population.

To account for preferential access to testing by HCWs during the early period of the pandemic, Fig 2 shows the positivity rate in HCWs as compared to the background population, suggesting that the high peak in HCW at the beginning of the pandemic shown in Fig 1 is

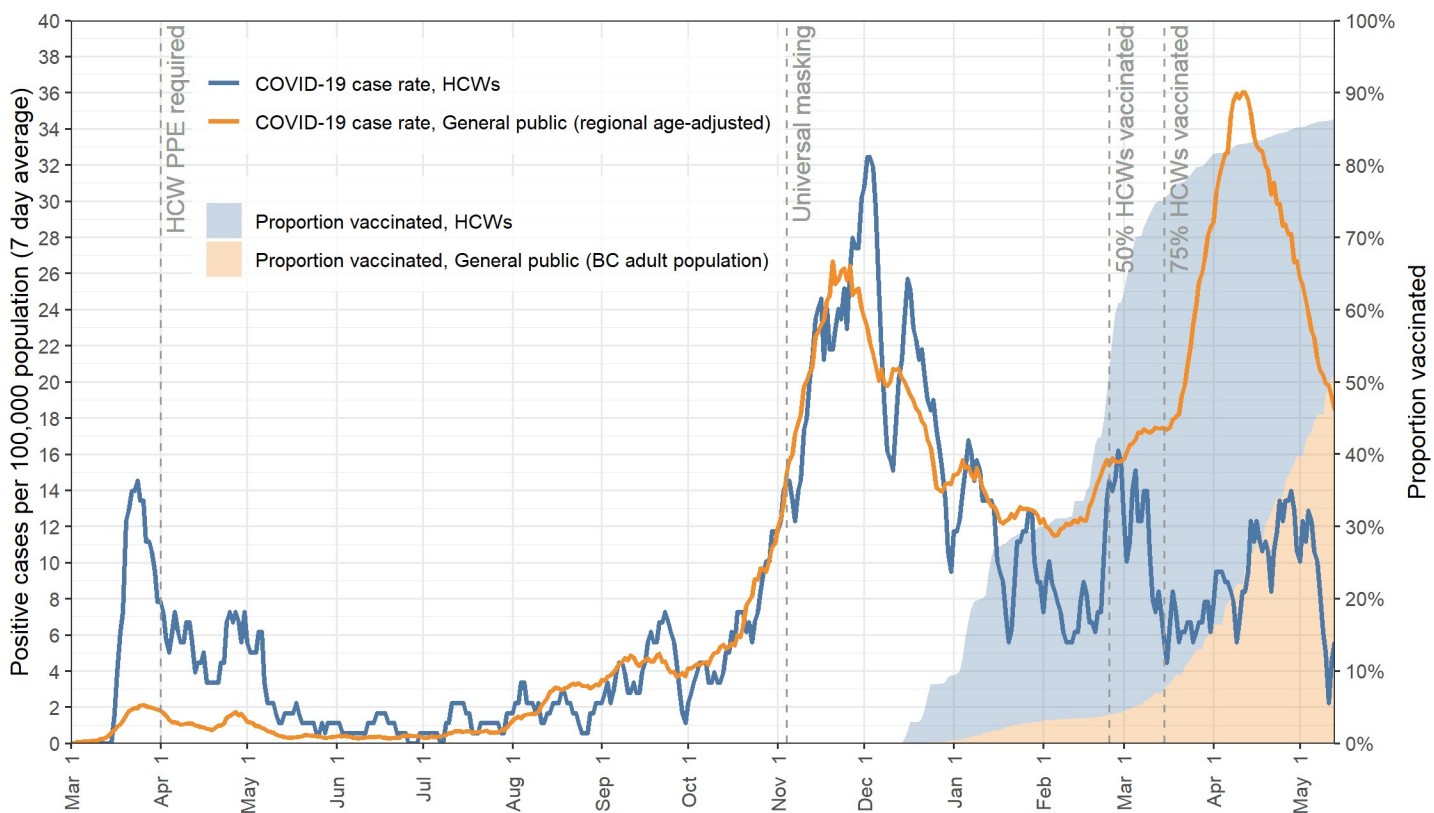

**Fig 1. COVID-19 incidence rates and vaccine coverage in healthcare workers of Vancouver Coastal Health compared to adjusted general population in area of residence over time.**

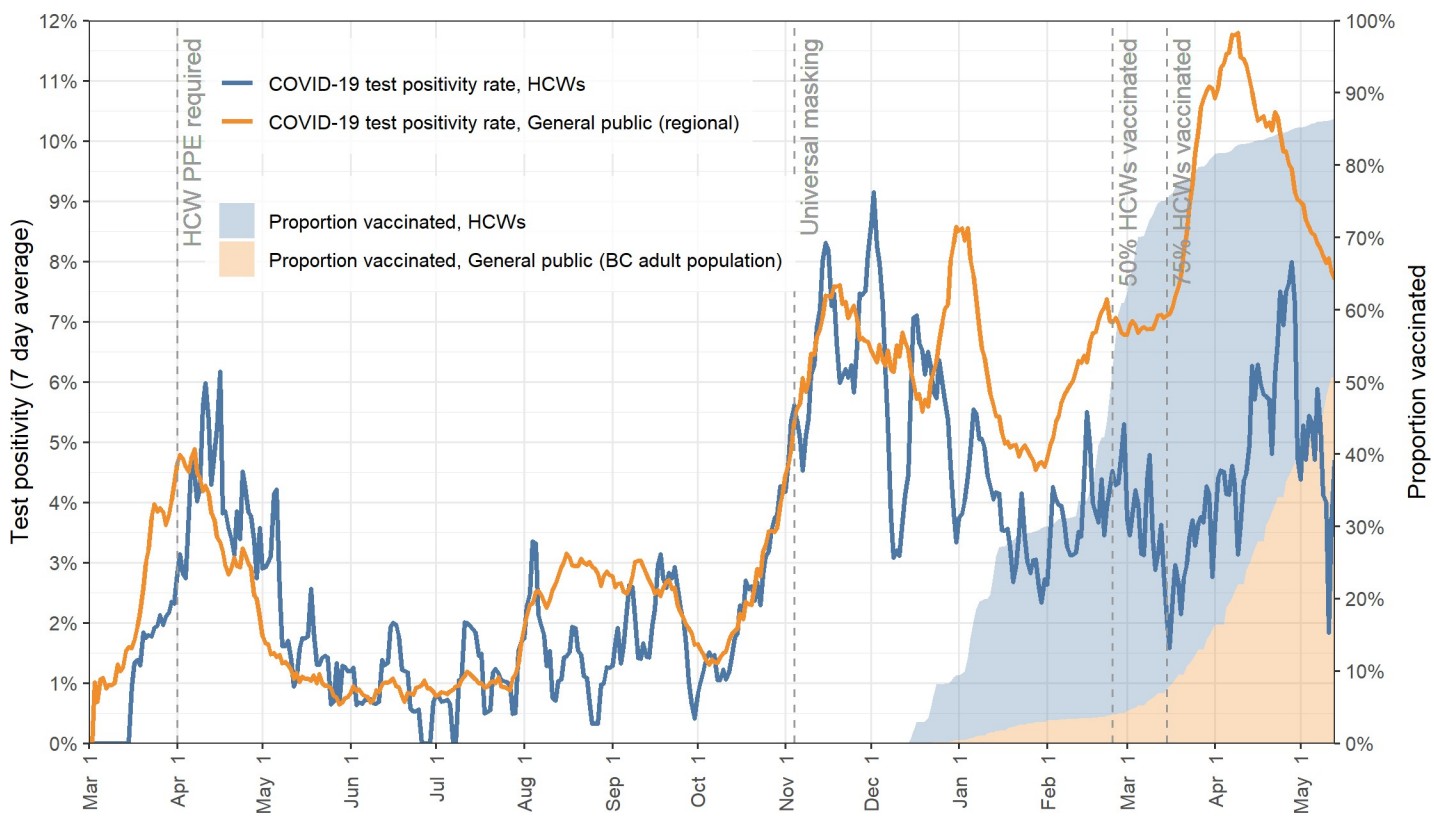

**Fig 2. COVID-19 positivity rate and vaccine coverage in in healthcare workers of Vancouver Coastal Health compared to general population in area of residence over time.**

likely due more to easier access of HCWs to COVID-19 testing. Positivity rates for our cohort of HCWs during the first wave (March 1- June 1, 2020) were 2.35% (95% CI, 1.94–2.84) while population rates were 2.27% (95% CI, 2.17–2.36). These figures similarly show a flat risk for HCWs, despite a community surge near the end of this observation period.

By the end of the observation period, 22,118 (86.5%) of HCWs had received at least one dose of vaccine, 7,328 (28.7%) had received two doses, with an average time between doses of 46.9 days (SD 18.7), leaving 3,440 (13.5%) unvaccinated. Table 1 shows COVID-19 rates and vaccine status for active employees, by occupation at the end of our observation period.

**Table 1. COVID-19 infection rate and vaccine status for VCH HCWs by occupation by May 13, 2021.**

| Occupation (n) | Cumulative COVID-19 rate n (%) | Fully vaccinated n (%) | Partially vaccinated n (%) | Unvaccinated n (%) |
|---|---|---|---|---|
| Nurses (7,637) | 247 (3.2%) | 2,274 (29.8%) | 4,433 (58.0%) | 930 (12.2%) |
| LPN/Care Aide (5,759) | 299 (5.2%) | 2,482 (43.1%) | 2,347 (40.8%) | 930 (16.1%) |
| Administration (4,314) | 114 (2.6%) | 405 (9.4%) | 3,137 (72.7%) | 772 (17.9%) |
| Allied Health (3,906) | 85 (2.2%) | 820 (21.0%) | 2,587 (66.2%) | 499 (12.8%) |
| Medical staff (3,182) | 68 (2.1%) | 1,353 (42.5%) | 1,700 (53.4%) | 129 (4.1%) |
| Support staff (827) | 35 (4.2%) | 120 (14.5%) | 548 (66.3%) | 159 (19.2%) |
| Other or Unknown (950) | 27 (2.8%) | 210 (22.1%) | 609 (64.1%) | 131 (13.8%) |
| Grand Total (25,558) | 837 (3.3%) | 7,328 (28.7%) | 14,790 (57.9%) | 3,440 (13.5%) |

*worked between March 15, 2020 –May 13, 2021. Excludes non-medical contract workers. An individual could have positions in multiple job classes, so can be counted in multiple rows. Grand total only counts each HCW once.

**Table 2. COVID-19 positive tests after vaccination in VCH HCWs from December 15, 2020.**

| When positive test occurred | Count | Days from last dose to positive test |
|---|---|---|
| Tested positive ≥14 days after first dose of vaccine | 98 (25.1%) | Median 47 days (IQR 33–61; Range 14–106)<br>Mean 48.1 days (95% CI 43.9–52.3) |
| Tested positive ≥7 days after second dose of vaccine | 16 (4.1%) | Median 54 days (IQR 44–62; Range 8–87)<br>Mean 53.1 days (95% CI 43.2–62.9) |
| Tested positive when unvaccinated or <14 days after first dose | 276 (70.8%) | -- |
| Before 1st dose | 220 (56.4%) | -- |
| < 14 days after first dose | 56 (14.4%) | Median 8 days (IQR 4–9; Range 1–13)<br>Mean 7.0 days (95% CI 6.1–7.9) |
| **TOTAL tested positive** | 390 (100.0%) | -- |

The crude cumulative population rate was 4.0%, or 4.4% when age-adjusted to match the workforce demographics, to allow comparison with our HCW cohort. By May 13, 2021, 3.3% of the VCH health workforce had tested positive for COVID-19, ranging from 3.2% in the acute sector to 4.6% in the long-term care sector; the highest risk occupational group was Licensed Practical Nurses and Care Aides (5.2%) and the lowest, medical staff (2.1%). During this time period, there was a shift from a rate of <1% VOC in early February to a high rate of two variants of concern (VOC) the B1.1.7 and P.1 variants–representing >92% of all infections by study closure, approximately evenly split between B.1.1.7 and P.1. [19].

There were 390 COVID-19 cases among active HCWs between Dec 15, 2020, and May 13, 2021. Of the 390 HCWs who became infected, 276 (70.8%) were unvaccinated or had received the first dose <14 days prior to their positive test, 98 (25.1%) tested positive 14 days or more after the first dose but before 7 days after the second dose, and 16 (4.1%) tested positive 7 days or more after the second dose (Table 2). Rates continued to decline despite climbing community rates.

The difference in COVID-19 rates between unvaccinated, vaccinated with one dose and fully vaccinated HCWs is shown in Table 3, where positive test results that occurred before 2 weeks after the first dose are counted in the unvaccinated category; similarly, positive tests that occur before 1 week after the 2nd dose are counted in the 1st dose category.

Rates of infection during the observation period were 1.33 per 10,000 person-days in unvaccinated HCWs, and 0.89 and 0.30 per 10,000 person-days for partially and fully vaccinated HCWs respectively (Table 3). This represents unadjusted reductions of COVID-19 of 33.2% (95% CI, 15.9% to 47.0%) and 77.6% (95% CI, 62.9% to 86.5%) for partially and fully vaccinated HCWs respectively. Compared with the unvaccinated community rates, unadjusted reductions were 54.7% (95% CI, 44.8% to 62.9%) and 84.8% (95% CI, 75.2% to 90.7%) for partially and fully vaccinated HCWs. These reductions are significant at 95% confidence, except for the interval comparing unvaccinated HCWs to those ≥42 days after the first dose, where reductions seem to be much smaller but the size of the population under observation is too small to make definitive statements in this regard. The Cox regression model, adjusted for age in years and calendar time, showed a reduction of COVID-19 infections of 37.2% (95% CI, 16.6 to 52.7%) ≥14 days after the first dose and 79.2% (95% CI, 64.6 to 87.8%) ≥7 days after the second dose. The cumulative infection rate of COVID-19 over time of unvaccinated, vaccinated ≥14 days with one dose and vaccinated ≥7 days with two doses is shown in Fig 3. A log-rank test shows that the incidence curves are significantly different (p < 0.001).

The vaccine effectiveness over time for partially and fully-vaccinated HCWs relative to the unvaccinated healthcare worker population, is shown in Fig 4.

**Table 3. COVID-19 positive tests by vaccination status over time in VCH HCWs and community (aged 20–69) from December 15, 2020 to May 13, 2021.** The adjusted rate is calculated using Cox regression, adjusting for age and calendar time.

| Group | Time range | Cases | Exposure person-days | Rate per 10,000 person- days | Unadjusted rate reduction compared with unvaccinated HCW | Adjusted rate reduction compared with unvaccinated HCW | Rate reduction compared with community rate |
|---|---|---|---|---|---|---|---|
| **Community (VCH & Fraser Health)** | December 15, 2020 to May 13, 2021 | 57,581 | $2.93 \times 10^8$ | 1.96 | -- | -- | -- |
| Effectively unvaccinated HCWs | Before 1st dose | 220 | 1,772,575 | 1.24 | -- | -- | -- |
| | < 14 days after 1st dose | 56 | 298,634 | 1.88 | -- | -- | -- |
| | Overall | 276 | 2,071,573 | 1.33 | -- | -- | -- |
| Partially vaccinated HCWs | 14–41 days after 1st dose | 39 | 578,496 | 0.67 | 49.4% (29.2% to 63.8%) | -- | 65.7% (53.0% to 74.9%) |
| | ≥ 42 days after 1st dose | 59 | 523,354 | 1.13 | 15.4% (-12.1% to 36.1%) | -- | 42.6% (25.9% to 55.5%) |
| | Overall | 98 | 1,101,850 | 0.89 | 33.2% (15.9% to 47.0%) | 37.2% (16.6% to 52.7%) | 54.7% (44.8% to 62.9%) |
| Fully vaccinated HCWs | ≥ 7 days after 2nd dose | 16 | 536,300 | 0.30 | 77.6% (62.9% to 86.5%) | 79.2% (64.6% to 87.8%) | 84.8% (75.2% to 90.7%) |

## Discussion

Protecting the healthcare workforce is an enduring priority. As such, ongoing vigilance on the adequacy of HCWs protection is essential, particularly in the face of growing concerns about

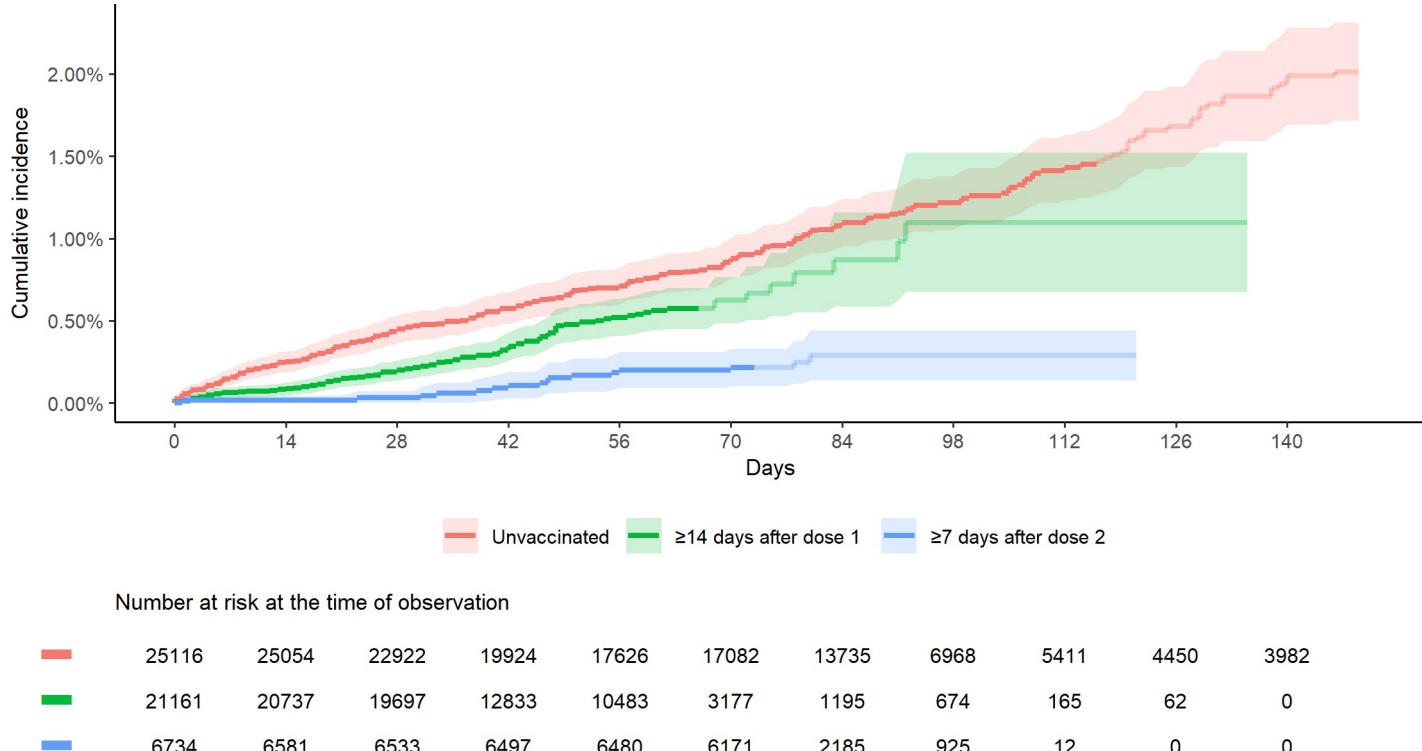

**Fig 3. Cumulative incidence of COVID-19 infection comparing unvaccinated, partially vaccinated and fully vaccinated healthcare workers of Vancouver Coastal Health over time.**

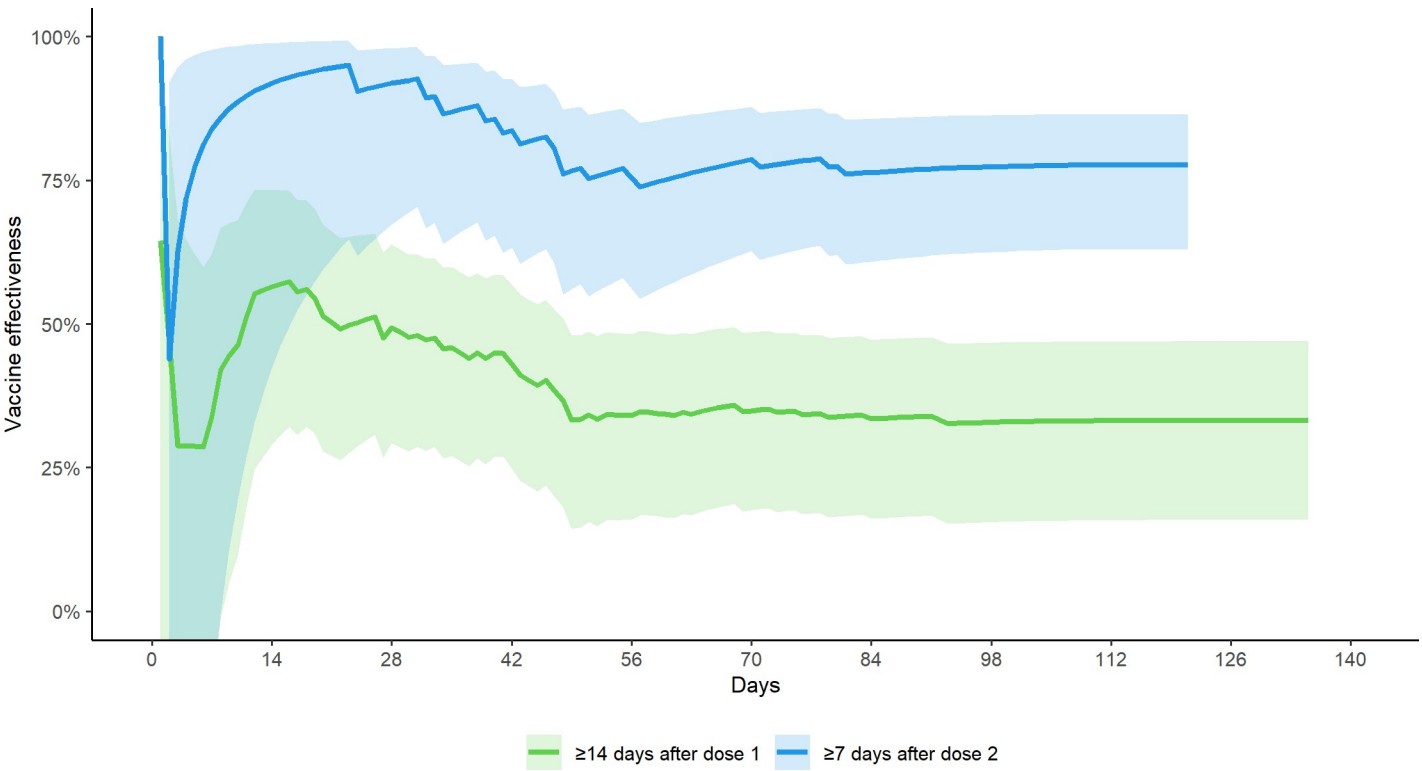

**Fig 4. Vaccine effectiveness (mRNA vaccine) comparing one dose and two doses over time.**

VOC and airborne transmission. Our data are consistent with the premise that the PPE recommendations in place (generally droplet-contact except where an AGMP is being performed) have provided good protection to workers, as part of a comprehensive rigorously implemented IPAC, Public Health and Occupational Health integrated program, with a permissive policy of N95 use based on a PCRA. While surveillance data lack details of which workers chose to wear an N95, or where they did so, this study has shown that PPE policies in place have worked well in our jurisdiction regardless of the proportion of viral transmission that may be airborne, and regardless of the high proportion of more transmissible variants.

While our relatively high rate of HCWs testing positive early in the pandemic may reflect a truly higher risk, it may be largely attributable to the selective testing strategy (due to limitations in testing capacity), which gave health workers preferential access to testing early in the pandemic. This hypothesis is supported by the observation that HCW positivity rates were similar to the background population. Increased case finding during outbreaks may also explain the differences in incident rates seen. Emecen and colleagues [20] showed that the serial interval and incubation periods of COVID-19 in HCWs were shorter than in the general population, which they suggest could be attributable to more rigorous contact tracing and isolation of infected HCWs [20].

Nonetheless, higher rates in HCWs with more extensive physical contact with patients (e.g. LPN, care aides) compared to others (administrators and medical staff) are concerning and may indeed suggest a role for occupational exposure in this group of workers. Case-control studies of risk factors for COVID-19 among HCWs [21,22] found that direct care to COVID-19 patients, unmasked close interaction with colleagues, and inappropriate use or shortage of PPEs were significant predictors of increased occupational risk among HCWs, along with

non-work-related risk factors. However, as suggested by others studies [23,24], the higher risk in this group of workers may also be explained by differences in socio-economic status and demographic factors which impacts variables such as private car use (versus public transit or carpooling), household composition, community of residence, and other important social determinants of health. We have a nested case-control study in progress to investigate the role of these putative risk factors in our cohort of VCH HCWs to supplement the rigorous analysis of surveillance data reported here. Regardless of whether the higher risk is predominantly attributable to community-based or workplace exposure, our data strongly indicate the need to prioritize these higher-risk workers in vaccination programs.

Our study shows 33.2% (95% CI, 15.9 to 47.0%) vaccine effectiveness against PCR-confirmed infections ≥14 days after first dose, and 77.6% (95% CI, 62.9 to 86.5%) effectiveness ≥7 days after the second dose when compared to unvaccinated HCWs, even with high rates of the B1.1.7 and P.1 variants. When compared to age-adjusted unvaccinated general public rates, we found vaccine effectiveness of 54.7% (95% CI, 44.8 to 62.9%) ≥14 days after the first dose, rising to 84.8% (95% CI, 75.2 to 90.7%) ≥7 days after the second dose. This represents a conservative estimate of vaccine effectiveness as infections that occur in the 14–21 period post first dose of vaccine may be due to infections acquired in the two weeks before antibodies developed. Moreover, the impact of vaccination on severity of disease was not captured in this analysis, and that infection rates are arguably of less concern than hospitalizations and no deaths occurred in our healthcare workforce.

Our findings are similar to other studies [18,25–27], which show vaccine-associated infection rate reductions following vaccination with increasing effectiveness from day 14 after first dose. For example, Dagan et al. [28] estimated vaccine effectiveness 14–20 days post first dose as 46% (95% CI, 40–51); and 92% (95% CI, 88–95) post second dose. Some studies showed higher rate reduction in vaccinated HCWs than our study [28,29]; this is likely explained by lower infection rates in our population: 1/10[th] of rates of HCW infected reported elsewhere (1.3 per 10,000 person-days in our cohort compared to 13.8 per 10,000 in a US cohort) [12] likely reflecting differences both in work and community transmission risk. Our lower infection rates in HCWs compared to studies in other settings add to the evidence that rigorous implementation of public health, occupational health and infection control measures can indeed keep HCWs protected at work and at home.

The higher rate of COVID-19 in the first two weeks following vaccination may be partly due to people dropping their guard thinking they are protected when antibodies have not yet developed, or individuals already infected and in the latency period at the time they were vaccinated. Likewise, because of changes in timing of a second dose the higher rate of infection in the period between the first and second doses, may be partially explained by the increasing community rates at the time longer-interval vaccine timing was occurring. Infections reached their nadir in mid-February, followed by an exponential rise in rates until mid-April, possibly partly accounting for the lower effectiveness reported for a single dose. This may bias against the efficacy of the single dose. Regardless, the need to ensure vigilance (at work and at home) in the 2-3-week period following vaccination is highlighted by our findings, as is the critical need for HCWs to receive two doses of vaccine to achieve excellent protection.

Unlike studies in other locations [30–32], in our jurisdiction fewer than 1% of all HCWs abstained from being vaccinated for medical or personal reasons, thus vaccine hesitancy is likely not a large issue. That higher risk occupational groups in our cohort–namely Licensed Practical Nurses and Care Aides–had slightly lower vaccination rates compared to lower risk healthcare worker groups is troubling. This may reflect less effective communication and outreach strategies to these workers compared to what nurses and medical staff receive, greater

difficulty in organizing vaccine appointments given work schedules, or greater vaccine hesitancy; further research is needed to understand the reasons for these differences.

Hall et al. [29] in their cohort of healthcare workers in England found their Pfizer vaccination was effective against the B1.1.7 strain, circulating at the time in the UK; their testing strategy in the UK was similar to ours with only symptomatic testing of HCWs conducted outside of outbreaks. Ours is the only report, of which we are aware, showing real-world effectiveness of vaccination in a population highly affected by the P.1 strain of SARS-CoV-2. While more study is needed, our results indicate that: 1) rigorous infection control measures have been effective in preventing occupational exposure; 2) vaccination has been effective in protecting HCWs from the impact of the third surge in which two VOC (B1.1.7 and P.1) predominated; 3) protection with one dose has been almost as effective as two doses for the first 42 days at least; and 4) the two-week period after vaccination is a high-risk period. The protection against infection gained from the single dose of the vaccine suggests that delaying administering the second dose to allow more people to have received at least one dose seems to have been well-founded; these findings provide strong support for guidelines suggesting an interval of at least up to 42 days between first and second dose [33]. A greater period of follow-up of our cohort is needed before more definitive statements can be drawn from this work regarding longer delays. Importantly, our study was not designed to assess real-world vaccine effectiveness related to severe disease, hospitalization and death; due to a high degree of occupational protection, we did not see widespread severe disease and death in our healthcare workforce even before the vaccination campaign began.

Overall, this study indicates that excellent protection can be achieved with predominantly droplet-contact infection prevention and control measures with N95s where appropriate, combined with prompt testing, tracing, isolation, and strong communication measures including with contracted medical staff, along with public health interventions that reduce pressure on the healthcare system. It further underlines the importance of the vaccination program, which we found to be effective in protecting HCWs from infection even in the face of VOC. That vaccination is far from perfect also underlines the need for ongoing vigilance in continued occupational protection, especially as variants continue to mutate and present new challenges. Nonetheless this study should provide some reassurance that the current approach of combining vaccination programs with infection prevention and control measures employed with a high degree of rigour, monitoring and communication, seems to indeed be effective to protect the healthcare workforce.

## Acknowledgments

The authors would like to sincerely thank all the healthcare workers who have worked tirelessly throughout the pandemic to take care of the ill and prevent transmission. For this study we recognize the contribution of the many University of British Columbia medical students who staffed the Physician Occupational Safety and Health (POSH) service, the Infection Prevention and Control, Public Health and People Safety teams who all worked to ensure that our healthcare workforce was well-protected, and the People Analytics team who extracted the data we needed for this research. We also express gratitude to the many others at Vancouver Coastal Health and within the province of BC who ensure excellent communications and a comprehensive surveillance system which guides operations and facilitates research.

## Author Contributions

**Conceptualization:** Annalee Yassi, Jennifer M. Grant.

**Formal analysis:** Stephen Barker.

**Funding acquisition:** Annalee Yassi.

**Investigation:** Annalee Yassi, Titus Wong.

**Methodology:** Annalee Yassi, Stephen Barker, Titus Wong.

**Project administration:** Karen Lockhart.

**Resources:** Jennifer M. Grant, Stacy Sprague, Titus Wong, Chad Kim Sing.

**Software:** Stephen Barker.

**Supervision:** Annalee Yassi, Karen Lockhart, Stacy Sprague, Arnold I. Okpani, Patricia Daly, Chad Kim Sing.

**Writing – original draft:** Annalee Yassi, Karen Lockhart.

**Writing – review & editing:** Annalee Yassi, Jennifer M. Grant, Stephen Barker, Stacy Sprague, Arnold I. Okpani, Titus Wong, Patricia Daly, William Henderson, Stan Lubin, Chad Kim Sing.

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
