## [Decision Letter · Decision Letter 0]

2 Jul 2021

PONE-D-21-16426

Infection control, occupational and public health measures including mRNA-based vaccination against SARS-CoV-2 infections to protect healthcare workers from variants of concern: a 14-month observational study using surveillance data

PLOS ONE

Dear Dr. Yassi,

Thank you for submitting your manuscript to PLOS ONE. After careful consideration, we feel that it has merit but does not fully meet PLOS ONE’s publication criteria as it currently stands. Therefore, we invite you to submit a revised version of the manuscript that addresses the points raised during the review process. In brief, the reviewer recommends minor revisions of the text.

We look forward to receiving your revised manuscript.

Kind regards,

David M. Ojcius

Academic Editor

PLOS ONE

Journal Requirements:

2. Please provide additional details regarding participant consent. In the ethics statement in the Methods and online submission information, please ensure that you have specified (i) whether consent was informed and (ii) what type you obtained (for instance, written or verbal, and if verbal, how it was documented and witnessed). If your study included minors, state whether you obtained consent from parents or guardians. If the need for consent was waived by the ethics committee, please include this information.

Reviewers' comments:

Reviewer's Responses to Questions

**Comments to the Author**

1. Is the manuscript technically sound, and do the data support the conclusions?

Reviewer #1: Yes

2. Has the statistical analysis been performed appropriately and rigorously? 

Reviewer #1: I Don't Know

3. Have the authors made all data underlying the findings in their manuscript fully available?

Reviewer #1: Yes

4. Is the manuscript presented in an intelligible fashion and written in standard English?

Reviewer #1: Yes

5. Review Comments to the Author

Reviewer #1: The authors Annalee Yassi et al have analysed surveillance data on the infectivity rate among healthcare workers in Vancouver, Canada. Their interpretation was 1) rigorous droplet contact precautions with N95 would prevent occupational infection 2) single dose mRNA vaccination further reduces risk of infection in healthcare workers and that delaying second dose allows widespread vaccination.

The study is quite relevant and interesting. However, I have one querry.

1. The vaccine effectiveness is only 37.2% in HCWs after single dose of mRNA vaccination when compared to unvaccinated HCWs for upto 42 days. They still have risk of contacting the infection. Hence the authors need to mention that second dose is also required for HCWs even if it is delayed.

2. Any idea about the Oxford (AstraZeneca) vaccine effectiveness in the context of HCWs under similar conditions as with mRNA vaccine?

6. PLOS authors have the option to publish the peer review history of their article (what does this mean?). If published, this will include your full peer review and any attached files.

Reviewer #1: No

---

## [Author Response · Author response to Decision Letter 0]

6 Jul 2021

Response to reviewers, PloS One

Yes, thanks. We have now formatted the titles, etc. to comply with these requirements as well as added acknowledgements and figure titles.

2. Please provide additional details regarding participant consent. In the ethics statement in the Methods and online submission information, please ensure that you have specified (i) whether consent was informed and (ii) what type you obtained (for instance, written or verbal, and if verbal, how it was documented and witnessed). If your study included minors, state whether you obtained consent from parents or guardians. If the need for consent was waived by the ethics committee, please include this information.

We have added a sentence re: consent.

Reviewers' comments:

Reviewer's Responses to Questions

Comments to the Author

1. Is the manuscript technically sound, and do the data support the conclusions?

Reviewer #1: Yes

Yes. We agree. However, as there was one question brought up when we published our pre-print that we wanted to address, we added an extra sentence in the methods and results, for completion.

2. Has the statistical analysis been performed appropriately and rigorously?

Reviewer #1: I Don't Know

3. Have the authors made all data underlying the findings in their manuscript fully available?

Reviewer #1: Yes

4. Is the manuscript presented in an intelligible fashion and written in standard English?

Reviewer #1: Yes

5. Review Comments to the Author

Reviewer #1: The authors Annalee Yassi et al have analysed surveillance data on the infectivity rate among healthcare workers in Vancouver, Canada. Their interpretation was 1) rigorous droplet contact precautions with N95 would prevent occupational infection 2) single dose mRNA vaccination further reduces risk of infection in healthcare workers and that delaying second dose allows widespread vaccination.

The study is quite relevant and interesting. However, I have one querry.

1. The vaccine effectiveness is only 37.2% in HCWs after single dose of mRNA vaccination when compared to unvaccinated HCWs for up to 42 days. They still have risk of contacting the infection. Hence the authors need to mention that second dose is also required for HCWs even if it is delayed.

Yes, thanks for highlighting this. We have now added a phrase to address the importance of the second dose.

2. Any idea about the Oxford (AstraZeneca) vaccine effectiveness in the context of HCWs under similar conditions as with mRNA vaccine?

We had very few HCWs in our cohort who received AstraZeneca (0.1%, 310 individuals). We have added a sentence to the article with this number.

6. PLOS authors have the option to publish the peer review history of their article (what does this mean?). If published, this will include your full peer review and any attached files.

Do you want your identity to be public for this peer review? For information about this choice, including consent withdrawal, please see our Privacy Policy.

Reviewer #1: No

---

## [Editor Report · Decision Letter 1]

7 Jul 2021

Infection control, occupational and public health measures including mRNA-based vaccination against SARS-CoV-2 infections to protect healthcare workers from variants of concern: a 14-month observational study using surveillance data

PONE-D-21-16426R1

Dear Dr. Yassi,

We’re pleased to inform you that your manuscript has been judged scientifically suitable for publication and will be formally accepted for publication once it meets all outstanding technical requirements.

Kind regards,

David M. Ojcius

Academic Editor

PLOS ONE

---

## [Editor Report · Acceptance letter]

9 Jul 2021

PONE-D-21-16426R1 

Infection control, occupational and public health measures including mRNA-based vaccination against SARS-CoV-2 infections to protect healthcare workers from variants of concern: a 14-month observational study using surveillance data 

Dear Dr. Yassi:

I'm pleased to inform you that your manuscript has been deemed suitable for publication in PLOS ONE. Congratulations! Your manuscript is now with our production department. 

Kind regards, 

on behalf of

Dr. David M. Ojcius 

Academic Editor

PLOS ONE